# Transcranial direct current simulation as an adjunctive treatment for treatment-resistant depression in hospitalized patients: A feasibility study protocol

John W. Li[1], Caili Ren[1¤], Vanessa K. Pazdernik[1], Simon Kung[1], Michael R. Basso[1], Paul E. Croarkin[1], Bina Aaron[2], Amanda M. Anderson[2], Courtney A. Caves[1], Misty N. Hemm[2], Ashley L. Holland[1], Charlie R. Hoth[1], Emily A. Jazdzewski[2], Eduardo Kabristante[2], Mahathi Kandimalla[3], Carly B. Mickle[2], Hoon-Ki Min[4,5,6], Can Ozger[1], Justine M. Parsons[2], Roberta L. Sheldon[2], Michelle K. Skime[1], Erik K. St. Louis[7], Yogatheesan Varatharajah[1], Neeraj Wagh[8], Kirk M. Welker[4], Sarah M. Williams[1], Gregory A. Worrell[7], Maria I. Lapid[1]*

1 Department of Psychiatry and Psychology, Mayo Clinic, Rochester, Minnesota, United States of America, 2 Department of Nursing, Mayo Clinic, Rochester, Minnesota, United States of America, 3 Department of Biomedical Sciences, Mayo Clinic, Rochester, Minnesota, United States of America, 4 Department of Radiology, Mayo Clinic, Rochester, Minnesota, United States of America, 5 Department of Biomedical Engineering, Mayo Clinic, Rochester, Minnesota, United States of America, 6 Department of Neurosurgery, Mayo Clinic, Rochester, Minnesota, United States of America, 7 Department of Neurology, Mayo Clinic, Rochester, Minnesota, United States of America, 8 Department of Bioengineering, University of Illinois Urbana-Champaign, Urbana, Illinois, United States of America

¤Department of Rehabilitation Medicine, Wuxi Central Rehabilitation Hospital, The affiliated Mental Health Center of Jiangnan University, Wuxi, Jiangsu, China

* lapid.maria@mayo.edu

## Abstract

Transcranial direct current stimulation (tDCS) is clinically effective in treating treatment-resistant depression (TRD), as measured by response, symptom improvement, and disease remission. However, the feasibility and underlying mechanism of tDCS treatment in individuals with TRD during acute psychiatric hospitalization remain poorly characterized. This paper outlines the protocol that aims to investigate the feasibility of implementing a 5-day tDCS treatment in hospitalized patients with TRD and secondarily explore the effects on depression and cognition, and neurophysiological mechanisms underlying tDCS. Current study will enroll ten participants who are diagnosed with TRD and are hospitalized in psychiatric units. Participants will receive a 5-day tDCS treatment protocol, with each treatment session lasting for 30 minutes, delivered twice daily, for a total of 10 stimulations over 5 days. The primary outcomes are the feasibility, acceptability, and tolerability of administering a 5-day tDCS treatment protocol in acutely hospitalized TRD patients. Exploratory outcomes pre- and post-tDCS include measures of depression (Montgomery-Asberg Depression Rating Scale (MADRS)) and cognition (Stroop Test, Revised Hopkins Verbal Learning Test (HVLT-R), Digital Symbol Coding Test (DSCT)), EEG changes

**Data availability statement:** Deidentified research data will be made publicly available when the study is completed and published.

**Funding:** This publication was made possible by the Mayo Clinic Center for Clinical and Translational Science (CTSA) through grant number UL1TR002377 from the National Center for Advancing Translational Sciences (NCATS), a component of the National Institutes of Health (NIH). Small grants administered under the CTSA funded our work. The funder did not play any role in the study design, data collection and analysis, decision to publish, or preparation of the manuscript.

**Competing interests:** This publication was made possible by the Mayo Clinic CTSA through grant number UL1TR002377 from the National Center for Advancing Translational Sciences (NCATS), a component of the National Institutes of Health (NIH). Functional near-infrared spectroscopy device is provided by OBELAB Inc. The sponsors did not play any role in the study design, data collection and analysis, decision to publish, or preparation of the manuscript.

in peak alpha frequency (PAF), and cerebral hemodynamic changes by functional near-infrared spectroscopy (fNIRS). This protocol would provide feasibility evidence for tDCS as an add-on to the standard of care treatment of TRD in hospitalized patients. Upon completion of the protocol, the preliminary effects of the 5-day tDCS treatment protocol regarding depression and cognitive symptoms and its neurophysiological mechanisms will be identified to guide the design and delivery of a randomized controlled study. Trial registration: National Institute of Health Clinicaltrials.gov (NCT06236711) and protocol ID: 23–003274.

## Introduction

Depression is a significant public health problem and one of the leading causes of disability worldwide. According to the National Institute of Mental Health report in 2020, an estimated 21 million adults in the United States had at least one episode of major depression in the past year, representing 8.4% of the adult population [1]. Globally, the World Health Organization reports that approximately 280 million people suffer from depression, constituting about 5% of the worldwide adult population [2]. Additionally, depression is a common co-morbidity associated with medical and neurologic disease [3,4]. While current pharmacologic and nonpharmacologic treatments for depression are available, there is a significant subset of patients who do not respond to the usual treatments. Treatment-resistant depression (TRD) is most commonly defined as a condition where an individual with major depressive disorder had failed a minimum of two or more treatment attempts that are considered adequate trials in terms of dosing and duration [5,6]. The prevalence of TRD varies greatly due to wide variations in definition, study populations, and inter-and intra-individual differences. Among adults treated for depression, the prevalence of TRD has been reported to range from 6 to 55% across different studies [7,8]. The impact of TRD on individuals, families, and societies is profound, with suicide being the most severe negative consequence.

In the context of the ongoing search for effective treatments for TRD, novel nonpharmacologic techniques, such as noninvasive brain stimulation (NIBS), have been more frequently used as primary or adjunctive treatments. NIBS involves stimulating specific areas of the neural circuitry, which are thought to regulate mood or other neuropsychiatric symptoms. The two most common forms of NIBS are 1) transcranial magnetic stimulation (TMS) which involves the use of electromagnetic fields and 2) transcranial direct current stimulation (tDCS) which involves the use of electric currents [9]. While TMS is already Food and Drug Administration (FDA)-approved for the treatment of TRD in the United States, tDCS remains investigational as a treatment for major depressive disorder and other neuropsychiatric conditions. To stimulate the brain with tDCS, a low-intensity electric current is delivered directly to the scalp through electrodes positioned on the head, held in place by a headband. The electric current is generated from a portable battery-operated device. tDCS changes the voltage of neuronal resting membrane potential toward depolarization after anodal

stimulation (excitatory) and toward hyperpolarization after cathodal stimulation (inhibitory). Alteration in neuronal excitability subsequently affects the likelihood of action potentials firing [10].

Based on a systematic review of tDCS for the treatment of depression, treatment parameters used across studies included ranges of 1–2 mA current intensity, 25–35 cm² electrode size, F3 (anodal) electrode position, and right supra-orbital (cathodal) reference electrode position. Treatment sessions occurred once or twice per day for 20–30 minutes, resulting in a total of 5–15 stimulations over a 1–3 weeks treatment duration [11]. Results of tDCS studies show inconsistent efficacy, primarily due to small sample sizes and heterogeneity of study populations. A recent meta-analysis demonstrated a modest effect of tDCS compared to sham (k = 25, Hedges's g = 0.46, 95% confidence interval [CI]: 0.22–0.70) in treating depressive episodes [12]. In reviewing a total of 14 randomized controlled trials and open-label studies conducted between 2012 and 2023, involving 967 subjects with a mean age 49.8 years (range from 18 and older), provides favorable evidence that tDCS is clinically effective in treating TRD [13]. The effectiveness is evaluated based on response rate, symptom improvement, and disease remission when compared to sham treatment or other treatment modalities, such as cognitive behavior therapy (CBT) and selective serotonin reuptake inhibitors (SSRI). Additionally, tDCS has shown to enhance cognition and may augment psychotherapy. However, most of the studies have been conducted in outpatient settings. Further research is needed to determine the treatment benefits for currently psychiatrically hospitalized TRD patients.

Some tDCS studies included TRD populations, but few studies have been conducted in inpatient settings. Patients hospitalized psychiatrically for TRD have high severity of symptoms, more failed prior treatments, and are at high risk for suicide and disability. Most of these patients feel they have tried enough medications and want to try a non-medication treatment approach. Hospitalization provides a window of opportunity to test whether adding tDCS as an adjunctive treatment to standard inpatient psychiatric care for TRD could be a viable option. In addition to offering patients a novel treatment, non-pharmacological treatments such as tDCS could also potentially decrease hospital length of stay. Therefore, we propose to test the feasibility, acceptability, and tolerability of using tDCS in the inpatient setting with TRD patients, explore electroencephalographic and cerebral hemodynamic change with tDCS, and assess its preliminary impact on both depressive and cognitive symptoms.

## Materials and methods

### Study design

The study is an open label feasibility study [14] designed to conduct a 5-day tDCS treatment protocol in 10 hospitalized adult patients with TRD, with each treatment session lasting for 30 minutes, delivered twice daily, for a total of 10 stimulations over 5 days [15,16]. Participants will receive neuropsychological assessments, electroencephalogram (EEG)

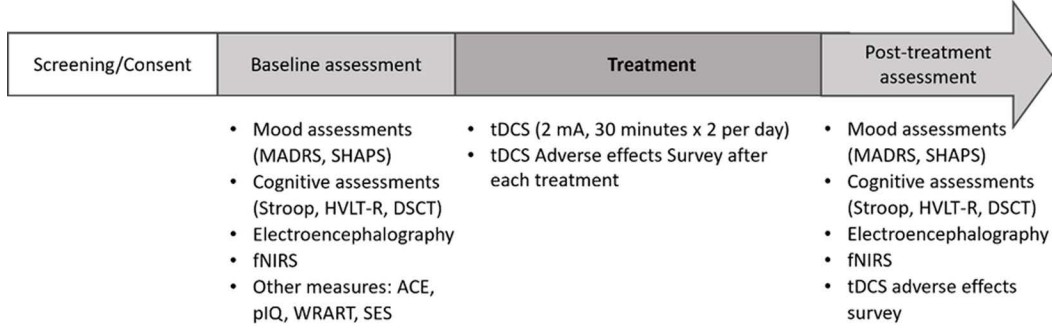

**Fig 1. Study timeline.**

examinations, and functional near-infrared spectroscopy (fNIRS) before and after the 5-day tDCS treatment protocol, as shown in Fig 1.

This study will be conducted at Mayo Clinic Rochester. Participants will be recruited from psychiatric inpatient units. Recruitment strategies will include study coordinators reviewing every patient admitted to the inpatient psychiatric units to pre-screen for eligibility. When a potential participant is identified, two psychiatrists will review the case for appropriateness, and then ask permission from the primary inpatient treatment team consultant or providers to approach the patient. With the primary service's permission, the study coordinator will explain the study to the patient, answer questions about the study, and enroll and obtain written informed consent from the participant. The patient recruitment period and expected date for the completion of intervention and evaluation are estimated to require about one year from the start of the recruitment which is expected to begin in December 2024.

**Inclusion criteria.** Participants will be included if they: (1) are aged 18 years and older; (2) have a clinical diagnosis of treatment-resistant depression, which is defined as depression that does not remit following two or more treatment attempts of an adequate dose and duration of a minimum duration of 4 weeks [6]; (3) are hospitalized in the Psychiatric Units at Mayo Clinic Rochester with voluntary admission status; (4) reach moderate or severe depression with a total score of at least 15 defined by PHQ-9; (5) can provide informed consent and adhere to protocol.

**Exclusion criteria.** Participants will be excluded if they have: (1) bipolar disorder; (2) active primary psychotic or substance use disorders (except nicotine dependence) within the past year; (3) any active neurological condition (including seizure disorder, traumatic brain injury, stroke); (4) contraindications to tDCS (including pacemaker, metallic implants in the head or neck [except orthodontic hardware] and irritative skin disease); (5) current pregnancy or positive urine pregnancy test (clinical); and (6) any previous or concurrent neuromodulation therapy (including electroconvulsive therapy (ECT), rTMS, deep brain stimulation (DBS), vagus nerve stimulation (VNS), or transcranial electrical stimulation (TES)) within the last 3 months.

The schedule for data collection, aligned with the Standard Protocol Items: Recommendations for Interventional Trial (SPIRIT) guidelines, is presented in Fig 2.

## Procedure

**tDCS intervention.** tDCS intervention will be performed using the Soterix Medical 1 × 1 CT Low-Intensity Transcranial Electrical Stimulator (Soterix Medical, Inc. (New York, NY, USA)). During the tDCS treatments, the subjects will remain in their room, be seated in a comfortable position, and be instructed to sit quietly with their eyes open throughout the procedures. The anode will be placed over the left dorsolateral prefrontal cortex (L-DLPFC) (corresponding F3 area according to International 10–20 electroencephalographic system), while the cathode will be placed over the right supraorbital area. The electrode size is $5 \times 7 \, cm^2$. tDCS is administered for 30 mins per session at 2.0 mA. Following the completion of one session, participants will undergo another session with a minimum inter-session interval of 30 minutes. A total of 2 tDCS sessions will be delivered per day. The 10 sessions will be applied over 5 consecutive weekdays. The operator can reduce current intensity if the participant experiences significant discomfort. The goal will be to ensure stimulation at the highest current intensity (up to 2 mA) that the participant finds comfortable. If the current intensity is reduced in one session, the operator will attempt to increase the current intensity back to 1 mA (or the maximum tolerated by the participant) in the following 1–2 sessions if feasible.

**Feasibility outcome measures.** The primary goal of this study is the feasibility, acceptability, and tolerability of administering a 5-day tDCS treatment protocol in acutely hospitalized TRD patients. Table 1 shows the feasibility outcomes and associated criteria.

***Feasibility*** - We define feasibility as 70% of eligible patients choosing to enroll in the study. Other feasibility parameters will include the willingness of clinicians to refer and help recruit patients, the number of eligible participants, the willingness of participants to enroll, the retention of patients, and the completeness of data collection.

| Study Activity | Screening | Baseline | Treatment | Post-treatment |
|---|---|---|---|---|
| **Timepoint** | **Day -7 to -1 (Visit -1)** | **Day -3 to -1 (Visit 0)** | **Day 1 to 5 (Visits 1-10)** | **Day 5-7 (Visits 11)** |
| **ENROLLMENT:** | | | | |
| **Inclusion/exclusion criteria** | X | | | |
| PHQ-9 | X | | | |
| Medical History | X | | | |
| Medication History | X | | | |
| Pregnancy Test (from EMR) | X | | | |
| **Informed Consent** | | X | | |
| Physical Exam | | X | | |
| **INTERVENTION:** | | | | |
| **tDCS treatment (2 mA, 30 min, twice a day)** | | | X | |
| **ASSESSMENTS:** | | | | |
| **EEG** | | X | | X |
| **fNIRS** | | X | | X |
| **Mood** | | | | |
| MADRS | | X | | X |
| SHAPS | | X | | X |
| **Cognition** | | | | |
| Stroop Test | | X | | X |
| HVLT-R | | X | | X |
| DSCT | | X | | X |
| **Other Assessments** | | | | |
| ACE | | X | | |
| pIQ | | X | | |
| SES | | X | | |
| WRART | | X | | |
| **Concurrent Medications** | | X | | |
| **Adverse Events** | | | X | X |
| **Serious Adverse Events** | | | X | X |

PHQ-9, Patient Health Questionnaire; EMR, electronic medical record; tDCS, transcranial direct current stimulation; EEG, electroencephalogram; fNIRS, functional near-infrared spectroscopy; MADRS, Montgomery-Asberg Depression Rating Scale; SHAPS, Snaith-Hamilton Pleasure Scale; HVLT-R, Revised Hopkins Verbal Learning Test; DSCT, Digital Symbol Coding Test; ACE, Adverse Childhood Experiences; pIQ, premorbid intelligence quotient; SES, socioeconomic status; WRART, Wide Range Achievement Reading Test.

**Fig 2. Summary of the proposed data collection schedule of enrolment, interventions, and assessment.**

**Acceptability** - We define acceptability as 80% of study participants completing 80% of the 10-session tDCS treatment protocol. We will also ask participants about their satisfaction, perceptions, and experiences with the tDCS treatments. Other acceptability parameters will include rates of adherence to tDCS protocol, dropout rates, and reasons for non-completion.

**Tolerability** - We define tolerability as an overall tolerability rating of "very tolerable." This will be a self-report based on a global question "Considering all aspects of your experience, how would you rate the overall tolerability of the tDCS

**Table 1. Feasibility outcomes and associated criteria.**

| Primary objective | Feasibility outcomes | Evaluation metrics | Analysis |
|---|---|---|---|
| Feasibility | Percentage of eligible patients to enroll in the study. | > 70% feasible | Descriptive |
| Acceptability | Percentage of participants completing 80% of the 10-session protocol. | > 80% feasible | |
| Tolerability | Self-report of participant's view on intervention. | overall tolerability rating of "very tolerable" by self-report | |

intervention?" The response options are very tolerable, somewhat tolerable, neutral, somewhat intolerable, and very intolerable. We will ask this question after the last tDCS session. Other tolerability parameters will include the tDCS adverse effects questionnaire.

## Exploratory outcome measures

**Neuropsychological measures.** Assessments will include the Montgomery-Asberg Depression Rating Scale (MADRS) to measure depressive symptoms [17], the Snaith-Hamilton Pleasure Scale (SHAPS) to measure anhedonia [18], the Stroop Test to measure memory and executive function, the Revised Hopkins Verbal Learning Test (HVLT-R) to test verbal learning and memory, and Digital Symbol Coding Test (DSCT) to measure working memory. Response is defined by a 50% or more reduction in the MADRS score [19,20]. Other potential predictors of treatment response measures collected at baseline include Adverse Childhood Experiences (ACE), premorbid intelligence quotient (pIQ), Wide Range Achievement Reading Test (WRART), and Socioeconomic Status (SES).

**EEG for electrophysiologic data.** A wireless EEG unit (CGX quick-20, Cognionic, Inc.) will be used to collect EEG and peak alpha frequency (PAF) as potential target engagement markers for tDCS. The PAF measures the highest magnitude within the alpha range of brain oscillations, thought to reflect cognitive performance [21]. The Quick-20r is a wireless, battery-operated EEG headset that uses dry sensor technology. It follows the international 10–20 system for electrode placement, which is based on specific measurements between landmarks on the head. The headset allows for the wireless acquisition of EEG signals, enabling subjects to move freely while data is collected in real-time. It provides high-quality EEG recordings with minimal scalp preparation. The device samples EEG channels at 500 Hz and converts the data to digital format with 24-bit resolution. The wireless EEG procedure will be carried out in 15–20 minutes, including a 5-minute eyes-open, a 5-minute eyes-closed, and device setup/removal time.

**Functional near-infrared spectroscopy (fNIRS) for cerebral hemodynamics.** Cerebral hemodynamics using fNIRS with the NIRSIT system (OBELAB Inc.) will be measured before and after tDCS. fNIRS indirectly measures cortical oxygenated and deoxygenated hemoglobin level ratio, providing insights into neural activation before and after brain stimulation. The NIRSIT device is a portable fNIRS device used in this study to transcutaneously measure changes in oxyhemoglobin in the prefrontal cortex using light detection. It is designed to measure variations in cerebral hemodynamics on a real-time basis by radiating a near light beam, at two wavelengths of 780nm and 850nm of laser, into the cerebral cortex. During the study, while the participant wearing the NIRSIT, changes in oxyhemoglobin are measured during the Verbal Fluency Test (VFT) [22,23]. The fNIRS acquisition will be carried out over 10–15 minutes including device setup, instructions, and completion of the VFT. The device instructions and VFT are automated processes using a pre-programed tablet device as a part of the NIRSIT system.

## Data collection methods

A Case Report Form (CRF) will be completed for each participant enrolled in the clinical study. The investigator will review, approve, and sign/date each completed CRF; the investigator's signature serves as an attestation of the investigator's responsibility for ensuring that all clinical data entered on the CRF are complete, accurate, and authentic.

The CRF is the primary data collection instrument for the study. All data requested on the CRF must be recorded. All missing data must be explained. The principal investigator (PI) will maintain records and essential documents related to the conduct of the study. These will include subject case histories and regulatory documents.

Data monitoring is conducted to assure data is accurate and complete. Monitoring of data assures adherence to the IRB-approved protocol. The PI and study team members will be responsible for data integrity. The PI will ensure participant inclusion criteria are met. The PI will provide oversight of entry of study data by data managers to ensure accuracy, and work with data managers in resolving any discrepancies in recorded or missing data.

Participants will be monitored throughout the entirety of all tDCS treatment sessions by study staff. This continuous monitoring will ensure that effective tDCS is delivered (e.g., good electrode contact).

Adherence to the treatment will be defined as completing a minimum of 8 treatments per week of the treatment course. If a participant misses 2 treatment session in a given week, another 2 treatment visits will be offered. If a participant misses more than two treatment sessions in a week, he or she will be withdrawn from the study. Should that occur, the participant will be offered the opportunity to participate in post-treatment assessments if they consent.

## Sample size

As this is a feasibility study, the selection of 10 patients was based on practical considerations such as resource availability. Future studies will include power calculations based on this study and references to prior research to support sample size decisions.

## Statistical analysis

Baseline values for demographic, clinical, and outcome variables (primary and secondary) will be reported as means and standard deviations for continuous data, or medians and interquartile ranges for skewed data, and frequencies and percentages for categorical data. The percentage of subjects completing the study and adhering to the protocol will be reported.

Paired *t*-tests or Wilcoxon signed rank tests will be used to compare pre- and post-tDCS measures on depression rating scales (MADRS, SHAPS), cognition (Stroop Test, HVLT-R, DSCT), PAF scores, and cerebral hemodynamic changes. The effect sizes reported will include the proportion that improved, and both raw mean and standardized mean differences, i.e., Cohen's d. Pearson or Spearman correlation will be used to estimate the correlation between MADRS scores and PAF at baseline, between MADRS scores and hemodynamic changes pre- and post-tDCS. Treatment response will be defined as 50% or more improvement in MADRS score post-treatment from pre-treatment. [19,20] In the context of exploratory analysis, associations between treatment responses with each baseline characteristic will be assessed using descriptive statistics and paired *t*-tests or Wilcoxon signed-rank tests.

For both the primary and secondary analyses, we will use the all-treated population and as a sensitivity analysis the set of subjects who completed at least 80% of 10 tDCS sessions. If missing data cannot be prevented, we will include cases with complete data for specific analyses. In the study report, we will be transparent about the extent of missing data and consider it as part of the study's overall success. We will apply appropriate methods, such as full information maximum likelihood estimation, to account for any remaining missing data in these analyses. The analysis will use SAS Studio 3.82 and R 4.3.2, with a significance level of 0.05 to reject the null hypothesis. However, the primary focus will be on descriptive statistics and effect size estimation, with a p-value interpretation considered secondary.

## Ethical considerations

**Ethics approval.** This study will be conducted according to United States government regulations and Institutional research policies and procedures. Ethics approval for this study was obtained from the Institutional Review Board (IRB)

of the Mayo Clinic (23–003274). All participants in this study will be provided a consent form describing this study and providing sufficient information for them to make an informed decision about their participation in this study. Written informed consent will be obtained from the initial visit.

**Adverse events and safety.** This is a greater than a minimal risk research study. Informed consent will be obtained from all participants before initiating study specific procedures in accordance with ethical principles, institutional policies and procedures, and federal regulations. The participant can withdraw at any time without affecting their clinical care. Adults 18 years and older are eligible to voluntarily participate in the study if they have capacity to consent to research.

Specific participant safety parameters for this study would include any adverse effects occurring while receiving tDCS. Based on prior research the main side effects could be redness of the skin or skin lesions, itching or scalp itching, scalp discomfort, tingling or burning sensations, sensation of a short light flash, dizziness, headache or pain, mood or sleep changes, nausea or fatigue.

Monitoring for adverse events will be conducted during the study period. Participants will be administered a tDCS Adverse Effects questionnaire after each treatment and at post-treatment assessment. The overall incidence of all adverse events (AEs) and serious AEs (SAEs), as well as the incidence of specific AEs and SAEs, will be determined.

## Discussion

This study seeks to evaluate the feasibility, acceptability, and tolerability of implementing a 5-day tDCS treatment protocol in acutely hospitalized patients diagnosed with TRD. The enhanced protocol, which involves twice-daily sessions, may lead to quicker symptoms relief for patients with TRD as it did for patients with MDD [24]. This approach may serve as a viable adjunctive treatment for hospitalized patients with TRD. The potential benefits include possible reduction of overall duration of hospitalization and extending the period of which the patients remain out of the hospital. Given the mild and transient nature of the adverse effects of tDCS as reported in the literature [25–27], the risks of the proposed tDCS treatment course appear to be outweighed by the potential benefit for improvement in depressive symptoms. Concurrently, this study will investigate the preliminary effects of tDCS on depressive symptoms and cognitive functioning, as well as explore electroencephalographic alterations and cerebral hemodynamic responses associated with tDCS interventions.

An innovative aspect of the proposed trial is the examination of the utility of tDCS in acutely psychiatrically hospitalized patients. In reviewing recent literature indicates that most RCTs utilizing tDCS for TRD have primarily targeted outpatient populations [13]. Therefore, demonstrating the feasibility of using tDCS in a hospital setting could provide additional treatment options for the challenging TRD population. Furthermore, the preliminary effects of a 5-day tDCS treatment protocol on depressive symptoms and cognitive status in individuals with TRD will be able to inform the methods for a future randomized trial of efficacy. The proposed neuropsychological test batteries, which include depressive outcome measures (MADRS, SHAPS) and cognitive measures (Stroop Test, HVLT-R, and DSCT), are well-established measures and each has been used in previous trials of tDCS to assess the treatment efficacy [11,12,15,28]. Lastly, neuroimaging measurements, such as PAF and changes in cerebral hemodynamics, may help elucidate the changes in brain activity patterns that underlie any physiologic changes in response tDCS.

In the context of TRD, the PAF has gained interest as a potential neurophysiological marker that provides insight into individual brain activity patterns that may influence responsiveness to noninvasive brain stimulation therapies such as tDCS. PAF represents the dominant frequency within the alpha band (typically between 8–12 Hz) where the alpha rhythms achieve maximum power [29]. This is measured via EEG and has been implicated in a variety of cognitive and affective processes [30]. In the context of TRD, there is a growing interest in the potential of PAF as a predictor or correlate of treatment response. Given that tDCS exerts its effects through modulating cortical excitability and neural plasticity, exploring correlations between PAF and tDCS responsiveness in TRD patients could offer valuable insights [28,31]. Capturing PAF data via EEG, therefore, may help elucidate the neurophysiological mechanisms underlying the therapeutic impact of tDCS.

The fNIRS technique has emerged as a non-invasive method for assessing brain activity function in recent years. Utilizing blood oxygen changes, fNIRS has been instrumental in discerning patterns of brain activity associated with conditions such as depression, bipolar disorder, and schizophrenia [32,33]. Notably, numerous studies have consistently reported decreased cerebral blood flow during the verbal fluency task (VFT) among individuals with Major Depressive Disorder (MDD) compared to their healthy counterparts [22,23]. Additionally, fNIRS investigations conducted immediately following rTMS, or after 10-day or 30-day course of rTMS treatment in patients with MDD, have revealed significant alterations in the temporal dynamics of oxyhemoglobin [34–37]. A recent randomized clinical trial protocol to assess combining aerobic exercise and tDCS on gait of patients with Parkinson's disease also highlights the utility of fNIRS to measure the cortical hemodynamic activity during tDCS treatments [38]. Thus, employing fNIRS to monitor cerebral hemodynamic changes during the VFT may offer insights into frontal activity responses to tDCS, potentially elucidating the underlying mechanisms of tDCS therapeutic efficacy.

A potential challenge to this study will be the small sample size and the enrollment and adherence of participants. The small sample size will be limited to the acutely hospitalized patients. These patients have a high severity of mood symptoms, are at high risk for suicide and disability. These factors may impede their willingness to enroll in an early phase clinical trial. On the other hand, although tDCS treatment appears well-tolerated, there will be some temporarily mild adverse events during the tDCS treatment that may affect the willingness of participants to adhere to tDCS during the trial.

In conclusion, this protocol would provide feasibility evidence for tDCS as an add-on to the standard of care treatment of TRD in hospitalized patients. Upon completion of the protocol, the preliminary effects of the 5-day tDCS treatment protocol regarding depression and cognitive symptoms and its neurophysiological mechanisms will be identified to guide the design and delivery of a randomized controlled study.

## Supporting information

**S1 File.  SPIRIT checklist.**
(PDF)

**S2 File.  Full study protocol.**
(PDF)

## Author contributions

**Conceptualization:** John W. Li, Caili Ren, Simon Kung, Michael R. Basso, Paul E. Croarkin, Hoon-Ki Min, Can Ozger, Erik K. St. Louis, Kirk M. Welker, Sarah M. Williams, Gregory A. Worrell, Maria I. Lapid.

**Data curation:** John W. Li, Caili Ren, Vanessa K. Pazdernik, Michael R. Basso, Hoon-Ki Min, Yogatheesan Varatharajah, Neeraj Wagh.

**Formal analysis:** John W. Li, Caili Ren, Vanessa K. Pazdernik, Michael R. Basso, Hoon-Ki Min, Yogatheesan Varatharajah, Neeraj Wagh.

**Funding acquisition:** John W. Li, Simon Kung, Paul E. Croarkin, Hoon-Ki Min, Maria I. Lapid.

**Investigation:** John W. Li, Caili Ren, Vanessa K. Pazdernik, Simon Kung, Michael R. Basso, Paul E. Croarkin, Bina Aaron, Amanda M. Anderson, Courtney A. Caves, Misty N. Hemm, Ashley L. Holland, Charlie R. Hoth, Emily A. Jazdzewski, Eduardo Kabristante, Mahathi Kandimalla, Carly B. Mickle, Hoon-Ki Min, Can Ozger, Justine M. Parsons, Roberta L. Sheldon, Michelle K. Skime, Erik K. St. Louis, Yogatheesan Varatharajah, Neeraj Wagh, Kirk M. Welker, Sarah M. Williams, Gregory A. Worrell, Maria I. Lapid.

**Methodology:** John W. Li, Caili Ren, Vanessa K. Pazdernik, Simon Kung, Michael R. Basso, Paul E. Croarkin, Bina Aaron, Amanda M. Anderson, Courtney A. Caves, Misty N. Hemm, Ashley L. Holland, Charlie R. Hoth, Emily A.

Jazdzewski, Eduardo Kabristante, Mahathi Kandimalla, Carly B. Mickle, Hoon-Ki Min, Can Ozger, Justine M. Parsons, Roberta L. Sheldon, Michelle K. Skime, Erik K. St. Louis, Yogatheesan Varatharajah, Neeraj Wagh, Kirk M. Welker, Sarah M. Williams, Gregory A. Worrell, Maria I. Lapid.

**Project administration:** John W. Li, Caili Ren, Simon Kung, Michael R. Basso, Courtney A. Caves, Misty N. Hemm, Charlie R. Hoth, Eduardo Kabristante, Mahathi Kandimalla, Carly B. Mickle, Hoon-Ki Min, Can Ozger, Justine M. Parsons, Roberta L. Sheldon, Michelle K. Skime, Yogatheesan Varatharajah, Neeraj Wagh, Sarah M. Williams, Maria I. Lapid.

**Resources:** John W. Li, Caili Ren, Michael R. Basso, Paul E. Croarkin, Emily A. Jazdzewski, Mahathi Kandimalla, Hoon-Ki Min, Can Ozger, Justine M. Parsons, Michelle K. Skime, Erik K. St. Louis, Yogatheesan Varatharajah, Neeraj Wagh, Kirk M. Welker, Sarah M. Williams, Gregory A. Worrell, Maria I. Lapid.

**Supervision:** Simon Kung, Michael R. Basso, Paul E. Croarkin, Hoon-Ki Min, Sarah M. Williams, Maria I. Lapid.

**Writing – original draft:** John W. Li, Caili Ren, Vanessa K. Pazdernik, Simon Kung, Michael R. Basso, Paul E. Croarkin, Hoon-Ki Min, Can Ozger, Erik K. St. Louis, Kirk M. Welker, Gregory A. Worrell, Maria I. Lapid.

**Writing – review & editing:** John W. Li, Caili Ren, Vanessa K. Pazdernik, Simon Kung, Michael R. Basso, Paul E. Croarkin, Bina Aaron, Amanda M. Anderson, Courtney A. Caves, Misty N. Hemm, Ashley L. Holland, Charlie R. Hoth, Emily A. Jazdzewski, Eduardo Kabristante, Mahathi Kandimalla, Carly B. Mickle, Hoon-Ki Min, Can Ozger, Justine M. Parsons, Roberta L. Sheldon, Michelle K. Skime, Erik K. St. Louis, Yogatheesan Varatharajah, Kirk M. Welker, Sarah M. Williams, Gregory A. Worrell, Maria I. Lapid.

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
