## [Decision Letter · Decision Letter 0]

22 Jul 2024

Dear Dr. Lapid,

Thank you for submitting your manuscript to PLOS ONE. After careful consideration, we feel that it has merit but does not fully meet PLOS ONE’s publication criteria as it currently stands. Therefore, we invite you to submit a revised version of the manuscript that addresses the points raised during the review process.

We look forward to receiving your revised manuscript.

Kind regards,

Abdolvahed Narmashiri

Academic Editor

PLOS ONE

Journal Requirements:

https://www.ajgponline.org/article/S1064-7481(24)00192-1/abstract

In your revision ensure you cite all your sources (including your own works), and quote or rephrase any duplicated text outside the methods section. Further consideration is dependent on these concerns being addressed.

**Additional Editor Comments:**

This protocol study provides evidence for the use of tDCS as an adjunct to standard care in the treatment of TRD in hospitalized patients. However, I have significant concerns regarding the methodological and scientific standards of this paper. A complete rewrite and reorganization are necessary to align it with accepted scientific standards.

1. Introduction: Remove the section titles and rewrite the introduction to present a coherent narrative.

2. Results: Add a dedicated results section to report pre- and post-tDCS outcomes, including measures of depression and cognition.

3. Methods: Rewrite the methods section to adhere to scientific standards.

4. Results Reporting: Ensure your results include:

o Depression measures (Montgomery-Asberg Depression Rating Scale (MADRS))

o Cognitive measures (Stroop Test, Revised Hopkins Verbal Learning Test (HVLT-R), Digital Symbol Coding Test (DSCT))

o EEG changes in peak alpha frequency (PAF)

o Cerebral hemodynamic changes (functional near-infrared spectroscopy (fNIRS))

o Attention to statistical analysis and accurate reporting in the results section.

5. Discussion: Remove the section titles and discuss the results in the context of the study protocol.

Reviewers' comments:

Reviewer's Responses to Questions

**Comments to the Author**

1. Does the manuscript provide a valid rationale for the proposed study, with clearly identified and justified research questions?

Reviewer #1: Yes

Reviewer #2: Partly

2. Is the protocol technically sound and planned in a manner that will lead to a meaningful outcome and allow testing the stated hypotheses?

Reviewer #1: Partly

Reviewer #2: Partly

3. Is the methodology feasible and described in sufficient detail to allow the work to be replicable?

Reviewer #1: Yes

Reviewer #2: Yes

4. Have the authors described where all data underlying the findings will be made available when the study is complete?

Reviewer #1: Yes

Reviewer #2: Yes

5. Is the manuscript presented in an intelligible fashion and written in standard English?

Reviewer #1: Yes

Reviewer #2: Yes

You may also provide optional suggestions and comments to authors that they might find helpful in planning their study.

**Reviewer #1: ** 1. This paper outlines a protocol to investigate the feasibility of implementing a 5-day tDCS treatment in hospitalized patients with TRD. The research also aims to explore the potential positive effects on depression and cognition and the neurophysiological mechanisms underlying tDCS. The authors did not indicate the specific descriptive statistics that will be used to summarize the data for continuous and categorical variables. They only mentioned descriptive statistics, which are too broad.

2. Since you are measuring the outcome measures before and after treatment for the same patients, the paired t-test or the non-parametric alternative will be more appropriate due to the correlation in the outcome measures.

3. Kindly state the statistical software that will be used for the analysis and at what level of significance the null hypothesis of no treatment effect will be rejected.

4. There was no justification for selecting 10 patients. There was no power analysis or sample size estimation to justify the choice of 10 and no reference to previous studies or expert documentation to justify the selection of 10 patients for the study.

5. Authors must provide references for the definition of feasibility outcomes. Some were set at >70% and others >80%. There must be references or justification to support the cut-off points or the 70% and 80% thresholds. Why not 90%, 50%, etc.? Kindly provide detailed justification or references to support the cut-off points.

6. Again, the authors indicated that treatment response will be defined as 50% or more improvement in MADRS score post-treatment from pre-treatment, but no justification was provided as to why they are setting the cut-off at 50% and not any other threshold. My concern is that setting the threshold at different cutoffs can generate entirely different results and conclusions, and that is more reason why justification must be provided

7. The authors anticipate the possibility of missing data but did not indicate which statistical technique will be used to handle missing values if it does occur. This is important as only 10 patients will be studied.

8. Complete case analysis, as suggested by the authors, in the presence of missing data reduces the sample size and the power of the study. This is particularly important when the authors have decided to study only 10 patients without details on how they arrived at the number 10 and the implications for missing observations.

9. Authors may re-consider running logistic regression on just 10 observations, with the possibility of a reduced sample size to an even smaller figure below 10.

**Reviewer #2:**  Title:

- Please do not use acronyms in the title.

Keywords:

- Please do not use the same words/expressions used in the title as keywords.

Ethics Statement:

- Please add information about the registration of the randomized clinical trial that will be carried out from the study protocol. Add the name and registration number to the Abstract session as well.

Introduction

- Line 76: What does “FDA” mean? Please write it in full before using the acronym.

- In general, I found the writing of the introduction to be exaggeratedly long. Information about TMS and fNIRS is not necessary here. I suggest placing more emphasis on the main focus of the study (effects of tDCS in TRD populations).

Methods

- Please add information about the sample calculation.

- Please add information about study blinding and randomization between active and sham.

- Please indicate when the participant recruitment period will begin and the expected date for the completion of interventions/evaluations.

- Explain the reason for applying tDCS twice a day.

- Please include references that support your interpretations for the “feasibility”, “acceptability” and “tolerability” measures.

- Please provide more details on the equipment used to measure cortical activity (EEG and fNIRS).

- Do you not intend to evaluate changes in deoxyhemoglobin? Please contextualize

Discussion

- This section needs to be better written, it is necessary to present information about protocols of other similar studies. Furthermore, I recommend making the innovative aspect of the study more evident.

**Do you want your identity to be public for this peer review?** For information about this choice, including consent withdrawal, please see our Privacy Policy

Reviewer #1: No

Reviewer #2: **Yes: ** Gabriel Antonio Gazziero Moraca

---

## [Author Response · Author response to Decision Letter 1]

11 Dec 2024

Reviewer #1:

1. This paper outlines a protocol to investigate the feasibility of implementing a 5-day tDCS treatment in hospitalized patients with TRD. The research also aims to explore the potential positive effects on depression and cognition and the neurophysiological mechanisms underlying tDCS. The authors did not indicate the specific descriptive statistics that will be used to summarize the data for continuous and categorical variables. They only mentioned descriptive statistics, which are too broad.

Response: Thank you. We have revised the sentence to be more specific as follows: “Baseline values for demographic, clinical, and outcome variables (primary and secondary) will be reported as means and standard deviations for continuous data, or medians and interquartile ranges for skewed data, and frequencies and percentages for categorical data.”

2. Since you are measuring the outcome measures before and after treatment for the same patients, the paired t-test or the non-parametric alternative will be more appropriate due to the correlation in the outcome measures.

Response: Thank you. We have clarified that the t-test used will indeed be paired. The Wilcoxon signed-rank test is the appropriate non-parametric alternative.

3. Kindly state the statistical software that will be used for the analysis and at what level of significance the null hypothesis of no treatment effect will be rejected.

Response: Thank you. We have added the following statement: “The analysis will use SAS Studio 3.82 and R 4.3.2, with a significance level of 0.05 to reject the null hypothesis. However, the primary focus will be on descriptive statistics and effect size estimation, with a p-value interpretation considered secondary.”

4. There was no justification for selecting 10 patients. There was no power analysis or sample size estimation to justify the choice of 10 and no reference to previous studies or expert documentation to justify the selection of 10 patients for the study.

Response: Thank you. We acknowledge the absence of a formal power analysis or sample size estimation. As this is a feasibility study, the selection of 10 patients was based on practical considerations such as resource availability. Future studies will include power calculations based on this study and references to prior research to support sample size decisions. We have added a Sample size section to the manuscript to clarify this.

5. Authors must provide references for the definition of feasibility outcomes. Some were set at >70% and others >80%. There must be references or justification to support the cut-off points or the 70% and 80% thresholds. Why not 90%, 50%, etc.? Kindly provide detailed justification or references to support the cut-off points.

Response: The cut-off points of >70% and >80% were selected based on common practices observed in similar feasibility studies and guidelines for early-phase research, where these thresholds often represent realistic and challenging benchmarks for assessing feasibility.

https://www.ncbi.nlm.nih.gov/pmc/articles/PMC8646008/ reports a target of <20% citing tDCS as reason for refusal to enroll and not quite similar is the target >10% enrolled out of those eligible.

6. Again, the authors indicated that treatment response will be defined as 50% or more improvement in MADRS score post-treatment from pre-treatment, but no justification was provided as to why they are setting the cut-off at 50% and not any other threshold. My concern is that setting the threshold at different cutoffs can generate entirely different results and conclusions, and that is more reason why justification must be provided

Response: Thank you. We have added two references (Leucht, Fennema et al. 2017, Turkoz, Alphs et al. 2021) supporting that MADRS total score of ≥50% corresponds to a clinically significant improvement such as correlation with the Clinical Global Impression Severity (CGI-S) score of 2 (i.e. much improved).

7. The authors anticipate the possibility of missing data but did not indicate which statistical technique will be used to handle missing values if it does occur. This is important as only 10 patients will be studied.

Response: Thank you. We have added that maximum likelihood estimation will be used in models with incomplete data to handle missing data. If missing data occurs, addressing this issue will become a primary focus regarding the feasibility study. More specifically we’ll be adhering to these guidelines depending on the scenario and data missing mechanism: https://statisticalhorizons.com/wp-content/uploads/MissingDataByML.pdf

8. Complete case analysis, as suggested by the authors, in the presence of missing data reduces the sample size and the power of the study. This is particularly important when the authors have decided to study only 10 patients without details on how they arrived at the number 10 and the implications for missing observations.

Response: We acknowledge this analysis may not be sufficiently powered to detect the true effects of this intervention on our measured outcomes. However, feasibility studies are not expected to have the large sample sizes that are needed to adequately power statistical null hypothesis testing. Indeed, pilot studies that are published often do not show statistically significant findings and rarely lead to larger trials to adequately power the hypothesis testing (Arain, Campbell et al. 2010, Shanyinde, Pickering et al. 2011). We have clarified this aspect of the study as documented in our response to comment #4.

9. Authors may re-consider running logistic regression on just 10 observations, with the possibility of a reduced sample size to an even smaller figure below 10.

Response: We acknowledge the concern regarding the small sample size for logistic regression. Given the limited number of observations, we recognize that the results may be unstable and lack power. We have changed to “…will be assessed using descriptive statistics and paired t-tests or Wilcoxon signed-rank tests.”

Reviewer #2:

Title:

- Please do not use acronyms in the title.

Response: Thank you. We have removed the acronyms.

Keywords:

- Please do not use the same words/expressions used in the title as keywords.

Response: Thank you. We have removed the duplicate words used in title.

Ethics Statement:

- Please add information about the registration of the randomized clinical trial that will be carried out from the study protocol. Add the name and registration number to the Abstract session as well.

Response: Thank you. We have added the name and registration number of the trial in the Abstract.

Introduction

- Line 76: What does “FDA” mean? Please write it in full before using the acronym.

Response: Thanks for pointing out the undefined acronym. FDA is now written in full “Food and Drug Administration.”

- In general, I found the writing of the introduction to be exaggeratedly long. Information about TMS and fNIRS is not necessary here. I suggest placing more emphasis on the main focus of the study (effects of tDCS in TRD populations).

Response: Thank you for the suggestions. We modified the introduction to be more concise and focused on the main objective of the study.

Methods

- Please add information about the sample calculation.

Response: Thank you. We acknowledge the absence of a formal power analysis or sample size estimation. As this is a feasibility study, the selection of 10 patients was based on practical considerations such as resource availability. Future studies will include power calculations based on this study and references to prior research to support sample size decisions. We have added a Sample size section to the manuscript to clarify this.

- Please add information about study blinding and randomization between active and sham.

Response: This is an open label study. Clarification is added in Study design section

- Please indicate when the participant recruitment period will begin and the expected date for the completion of interventions/evaluations.

Response: We expect patient recruitment to start in December 2024 and all interventions and evaluations will be completed about one year from the start of the recruitment.

- Explain the reason for applying tDCS twice a day.

Response: The reason for applying tDCS twice per day over 5 consecutive days is to maximize the number of treatments over a typical course of inpatient stay, which is on average 5 to 7 days long in our facility. Previous studies have shown that it is safe to administer 10 sessions of 30 minutes tDCS over 5 days in depressed patients and that two daily tDCS sessions were potentially two daily tDCS sessions were potentially more effective in reducing depressive symptoms when compared to a single session per day (Bennabi, Nicolier et al. 2015, Zanardi, Poletti et al. 2020).

- Please include references that support your interpretations for the “feasibility”, “acceptability” and “tolerability” measures.

Response: The cut-off points of >70% and >80% were selected based on common practices observed in similar feasibility studies and guidelines for early-phase research, where these thresholds often represent realistic and challenging benchmarks for assessing feasibility.

https://www.ncbi.nlm.nih.gov/pmc/articles/PMC8646008/ reports a target of <20% citing tDCS as reason for refusal to enroll and not quite similar is the target >10% enrolled out of those eligible.

- Please provide more details on the equipment used to measure cortical activity (EEG and fNIRS).

Response: Thank you. We have modified the method section to include more details about the equipment and procedures for the EEG and fNIRS acquisition.

- Do you not intend to evaluate changes in deoxyhemoglobin? Please contextualize

Response: We are not planning on examine the changes in deoxyhemoglobin in the current study. It would one of our goals for a larger trial once the feasibility of acquiring fNIRS data during tDCS treatment of TRD in the inpatient environment.

Discussion

- This section needs to be better written, it is necessary to present information about protocols of other similar studies. Furthermore, I recommend making the innovative aspect of the study more evident.

Response: Thank you. We have modified the discussion section as suggested.

Editor comments

1. Introduction: Remove the section titles and rewrite the introduction to present a coherent narrative.

Response: We have removed the section titles and modified the introduction.

2. Results: Add a dedicated results section to report pre- and post-tDCS outcomes, including measures of depression and cognition.

Response: We do not have results to report with this manuscript as this is a proposal for a clinical trial.

3. Methods: Rewrite the methods section to adhere to scientific standards.

Response: We have modified the methods section.

4. Results Reporting: Ensure your results include:

- Depression measures (Montgomery-Asberg Depression Rating Scale (MADRS))

- Cognitive measures (Stroop Test, Revised Hopkins Verbal Learning Test (HVLT-R), Digital Symbol Coding Test (DSCT))

- EEG changes in peak alpha frequency (PAF)

- Cerebral hemodynamic changes (functional near-infrared spectroscopy (fNIRS))

- Attention to statistical analysis and accurate reporting in the results section.

Response: We do not have results to report with this manuscript as this is a proposal for a clinical trial. Currently, we are expecting patient recruitment to start soon, and all interventions and measurements above will be acquired then.

5. Discussion: Remove the section titles and discuss the results in the context of the study protocol.

Response: We have removed the section titles and modified the discussion section. We are not able to discuss any results at this time as we are currently in the process of starting the recruitment.

Journal Requirements:

Please ensure that your manuscript meets PLOS ONE's style requirements, including those for file naming. The PLOS ONE style templates can be found at…

Response: We have modified the manuscript to meet the PLOS ONE’s style requirements baed on the guidelines.

https://www.ajgponline.org/article/S1064-7481(24)00192-1/abstract

In your revision ensure you cite all your sources (including your own works), and quote or rephrase any duplicated text outside the methods section. Further consideration is dependent on these concerns being addressed.

Response: Thank you for pointing out the overlapping with our previous publication. We have updated the citation in the manuscript and made modifications to the text.

Response: Thank you. We have addressed this issue in the manuscript.

Response: Thank you for the explanation regarding open data policy.

Response: Thank you. We have modfied the manuscript to include the ethics statement only in the Methods section.

References

Arain, M., et al. (2010). "What is a pilot or feasibility study? A review of current practice and editorial policy." BMC Medical Research Methodology 10(1): 67.

Bennabi, D., et al. (2015). "Pilot study of feasibility of the effect of treatment with tDCS in patients suffering from treatment-resistant depression treated with escitalopram." Clin Neurophysiol 126(6): 1185-1189.

Leucht, S., et al. (2017). "What does the MADRS mean? Equipercentile linking with the CGI using a company database of mirtazapine studies." J Affect Disord 210: 287-293.

Shanyinde, M., et al. (2011). "Questions asked and answered in pilot and feasibility randomized controlled trials." BMC Medical Research Methodology 11(1): 117.

Turkoz, I., et al. (2021). "Clinically meaningful changes on depressive symptom measures and patient-reported outcomes in patients with treatment-resistant depression." Acta Psychiatr Scand 143(3): 253-263.

Zanardi, R., et al. (2020). "Transcranial direct current stimulation: A novel approach in the treatment of vascular depression." Brain Stimul 13(6): 1559-1565.

---

## [Decision Letter · Decision Letter 1]

2 May 2025

Transcranial direct current simulation as an adjunctive treatment for treatment-resistant depression in hospitalized patients: A feasibility study protocol

PONE-D-24-24817R1

Dear Dr. Lapid,

We’re pleased to inform you that your manuscript has been judged scientifically suitable for publication and will be formally accepted for publication once it meets all outstanding technical requirements.

Kind regards,

Abdolvahed Narmashiri

Academic Editor

PLOS ONE

Additional Editor Comments (optional):

Reviewers' comments:

Reviewer's Responses to Questions

**Comments to the Author**

1. Does the manuscript provide a valid rationale for the proposed study, with clearly identified and justified research questions?

Reviewer #1: Yes

2. Is the protocol technically sound and planned in a manner that will lead to a meaningful outcome and allow testing the stated hypotheses?

Reviewer #1: Yes

3. Is the methodology feasible and described in sufficient detail to allow the work to be replicable?

Reviewer #1: Yes

4. Have the authors described where all data underlying the findings will be made available when the study is complete?

Reviewer #1: Yes

5. Is the manuscript presented in an intelligible fashion and written in standard English?

Reviewer #1: Yes

You may also provide optional suggestions and comments to authors that they might find helpful in planning their study.

Reviewer #1: The authors have addressed all my concerns from the previous review. The manuscript has significantly improved

**Do you want your identity to be public for this peer review?** For information about this choice, including consent withdrawal, please see our Privacy Policy

Reviewer #1: No

---

## [Editor Report · Acceptance letter]

PONE-D-24-24817R1

PLOS ONE

Dear Dr. Lapid,

I'm pleased to inform you that your manuscript has been deemed suitable for publication in PLOS ONE. Congratulations! Your manuscript is now being handed over to our production team.

Kind regards,

on behalf of

Dr. Abdolvahed Narmashiri

Academic Editor

PLOS ONE